# Eating Styles Profiles and Correlates in Chinese Postpartum Women: A Latent Profile Analysis

**DOI:** 10.3390/nu16142299

**Published:** 2024-07-17

**Authors:** Jiayuan Peng, Tian Xu, Xiangmin Tan, Yuqing He, Yi Zeng, Jingfei Tang, Mei Sun

**Affiliations:** 1Xiangya School of Nursing, Central South University, No.172 Tongzipo Road, Yuelu District, Changsha 410013, China; pjyxx2022@163.com (J.P.); xutian927@163.com (T.X.); heyuqing299@126.com (Y.H.); zyxiangya@163.com (Y.Z.); tangjingfei853@163.com (J.T.); 2School of Rural Health, Monash University, 15 Sargeant Street, Warragul, VIC 3820, Australia; xiangmin_456@outlook.com; 3School of Nursing, Xinjiang Medical University, No.168 Youyi South Road, Urumqi 830054, China

**Keywords:** disordered eating behaviors, eating styles, latent profile analysis, postpartum women, person-centered approach

## Abstract

Postpartum women present a high risk of disordered eating behaviors, but the heterogeneity between groups was not identified. This cross-sectional study aimed to identify eating styles profiles in postpartum women and explore the correlates based on demographic characteristics and psychosocial factors. Questionnaires were administered to 507 Chinese postpartum women. Latent profile analysis (LPA) was conducted to identify eating styles profiles. Multinomial logistic regression was used to investigate the correlates of these profiles among postpartum women. The LPA identified three eating styles profiles: postpartum women with low emotional, external, and restrained eating (Profile 1, 6.9%); postpartum women with medium emotional, external, and restrained eating (Profile 2, 66.1%); and postpartum women with high emotional, external, and restrained eating (Profile 3, 27.0%). Compared to Profile 1, higher postpartum depression (PPD) and body mass index (BMI) were more likely to be associated with Profile 2 and Profile 3, whereas higher postpartum weight retention (PPWR) was more likely to be associated with Profile 1. Compared to Profile 2, higher PPD and BMI were more likely associated with Profile 3. Disordered eating behaviors in postpartum women with three eating styles were associated with BMI, PPD, and PPWR. This study can guide healthcare professionals in developing targeted interventions to improve maternal and child health globally.

## 1. Introduction

Overweight and obesity have evolved into a serious global public health issue. According to a report by the World Health Organization, 39% of adults were overweight and 13% were obese globally in 2016 [1]. It was predicted that, by 2030, 1 in 5 women and 1 in 7 men will be obese [2]. In this global obesity crisis, women of reproductive age are the highest risk group for accelerating the development of obesity. Global trends in obesity showed that women of reproductive age have the highest rate of increase in overweight and obesity among all age groups of men and women [3].

The development of overweight and obesity is influenced by multiple risk factors, including genetic, sociocultural, consumption of ultra-processed foods, psychosocial, and behavioral factors [4]. Among these factors, eating behaviors need particular attention [5]. Disordered eating behaviors have been recognized as an important and persistent risk factor for overweight and obesity [6]. Martin-Biggers et al. [7] surveyed 550 non-obese mothers and discovered a positive association between a high risk of obesity and emotional eating. Furthermore, Kessler et al. [8] observed that individuals with binge-eating disorder were three to six times more likely to be obese than those without disordered eating behaviors.

The Dutch Eating Behavior Questionnaire (DEBQ) has been widely utilized to assess three primary eating behaviors: emotional eating, external eating, and restrained eating [9]. These styles are rooted in three predominant psychological theories on disordered eating: psychosomatic theory, externality theory, and restraint theory. The psychosomatic theory [10] posits that emotional eating is an atypical response to distress, characterized by eating in response to negative emotions. The externality theory [11] describes external eating as a response to external food-related stimuli, such as the sight, smell, and taste of food, regardless of internal hunger and satiety cues. The restraint theory [12] focuses on dietary restraint and the potential psychological consequences of dieting, particularly the disinhibition effect, which leads to overeating when dieters abandon their cognitive control over eating less. These theories highlight the complex interplay of eating behaviors, suggesting that disordered eating can be driven by emotional external cues or cognitive restraint. The DEBQ’s inclusion of these three eating styles provides a robust framework for examining disordered eating behaviors.

Studies have found that women face a greater risk of disordered eating behaviors than men. The higher the body mass index (BMI), the greater the risk [13,14,15]. It is particularly noteworthy that the postpartum period constitutes a critical high-risk window for developing disordered eating behaviors in women. During this stage, physiological changes rendered women susceptible to excessive weight gain or retention [16]. Moreover, in the special sociocultural context of China, postpartum women are expected to engage in ‘confinement in childbirth’ during the puerperal period, which is characterized by reduced activity and the consumption of various nutritious soups [17]. These factors contribute to weight fluctuations in postpartum women. Meanwhile, women become more concerned about their body image and weight after delivery than before pregnancy [18]. More importantly, the transition to motherhood proves challenging, and disruptions in routine and sleep make it difficult for postpartum women to maintain regular eating patterns [19]. Additionally, women are less motivated to maintain a healthy lifestyle after delivery compared to during pregnancy [18,20]. These factors, interacting with the multiple stressors of motherhood, may distort perceptions of postpartum women, thereby contributing to a high risk of disordered eating behaviors [21].

Disordered eating behaviors were associated with negative health outcomes in postpartum women, including postpartum weight retention (PPWR) [22], postpartum depression (PPD) [23], and suicidal tendencies [24]. Furthermore, significant family implications exist, with partners and children facing a heightened risk of developing unhealthy eating behaviors [25,26]. Considering the impact of eating behaviors on postpartum women and the well-being of their families, it is imperative to clearly understand the unique patterns of disordered eating behaviors during this period, as well as the associated demographic and psychosocial characteristics.

In addition to BMI and gender, researchers have examined other demographic characteristics, yet they have not reached consistent conclusions. Regarding age, several studies suggested that emotional and external eating decreased with age [27,28], while restrained eating tended to increase [28]. However, other studies have shown no association between age and disordered eating behaviors [28]. In terms of social status, Bojorquez [29] observed that employment, higher income, and higher education levels were associated with lower levels of disordered eating behaviors. Yet, Barrada et al. [28] found no relationship between education level and disordered eating behaviors. 

Psychosocial factors have also been identified as important correlates of disordered eating behaviors. Extensive research on psychosocial factors provides empirical evidence for developing precise and effective interventions. Research has demonstrated positive correlations between depression, weight stigma, and disordered eating behaviors in postpartum women. A study involving 711 postpartum women in the United States showed that higher levels of depression were positively associated with increased emotional and restrained eating [30]. Another study, including 501 pregnant and postpartum women in the United States, found that women experiencing more weight stigma reported higher levels of emotional and restrained eating [31]. To the best of our knowledge, the existing studies on disordered eating behaviors in postpartum women have not sufficiently considered the critical factor of internalized weight stigma. However, the results from a systematic review suggested that internalized weight stigma played a crucial mediating role in the relationship between weight stigma and disordered eating behaviors [32].

While previous studies have demonstrated significant relationships between demographic characteristics, psychosocial factors, and eating behaviors, most have used a ‘variable-centered’ approach [33]. This method assesses disordered eating behaviors based on the total scale score, primarily examining relationships and interactions between variables. However, this approach limits the ability to identify within-group heterogeneity. To develop more targeted interventions, latent profile analysis (LPA) is a more suitable method for assessing disordered eating behavior characteristics within groups [34]. This ‘person-centered’ approach identifies an individual’s latent characteristics based on their responses to each item, classifying individuals into distinct profiles to better understand within-group heterogeneity [35]. LPA focuses on combinations of characteristics at the individual level and how these characteristics aggregate to form unique groups. Although several studies have used LPA to assess disordered eating behaviors in the general population [34,36], to date, no studies have applied LPA to analyze disordered eating characteristics in postpartum women, which is a high-risk population.

Therefore, the present study aimed to (a) identify eating styles profiles of postpartum women based on their levels of emotional eating, external eating, and restrained eating, and (b) explore the correlates based on demographic characteristics (e.g., BMI and PPWR) and psychosocial factors (e.g., PPD and weight stigma).

## 2. Materials and Methods

### 2.1. Participants

This cross-sectional study was conducted between December 2021 and March 2022 in the postnatal follow-up clinics of a maternal and child health hospital and four community health centers in China. A convenience sampling method was used to recruit postpartum women in the waiting room. Before distributing the questionnaires, the researchers explained the study’s purpose, risks, and benefits to the eligible participants. As compensation for their time, a bottle of milk or a CNY 5 red packet was given to each participant who volunteered. Paper questionnaires were distributed to all those who met the inclusion criteria and were willing to participate. Trained research staff guided participants in completing the questionnaires in a demonstration room and collected them on-site upon completion. The questionnaire could be completed in 10–15 min.

The inclusion criteria for participants were as follows: (a) at least 18 years of age; (b) within one year of childbirth; and (c) voluntary participation in this study. Individuals with a history of major physical problems or mental disorders that might affect cognitive function were excluded.

A previous study has suggested a reasonable minimum sample size for latent profile analysis should be approximately 500 [37]. Ultimately, 507 postpartum women were included in the analysis. The participants’ heights, weights, and BMIs were measured on-site.

### 2.2. Measures

#### 2.2.1. Demographic Characteristics

The socio-demographic data that were collected included age, residence, education level, monthly income, employment status, body mass index (BMI), postpartum weight retention (PPWR), and sleep condition.

#### 2.2.2. Perceived Weight Stigma Questionnaire (PWSQ)

The PWSQ is a self-reported questionnaire that assesses perceptions of weight-stigma experiences. The scale uses 10 dichotomous items, each scoring as ‘yes’ or ‘no’ (‘yes’ = 1, ‘no’ = 0). The items are averaged into an overall score, with higher scores reflecting more frequent weight-teasing experiences. The Chinese version of the scale was used to examine perceived weight stigma in university students and demonstrated good psychometric properties [38]. In this study, Cronbach’s α for the PWSQ was 0.80, showing adequate reliability and validity.

#### 2.2.3. Weight Bias Internalization Scale (WBIS)

The WBIS was developed by Durso et al. in 2008 [39] to measure the level of internalized weight stigma in the general population. The WBIS comprises 11 items rated on a 5-point Likert scale, ranging from 1 (strongly disagree) to 5 (strongly agree). Higher scores on the WBIS indicate a greater degree of internalized weight stigma. The WBIS has demonstrated satisfactory internal consistency (Cronbach’s α = 0. 87) among German adolescents who are overweight or obese [40]. The scale has also been extensively validated in Chinese children and adolescents, showing good internal consistency [41]. In this sample, the WBIS exhibited a Cronbach’s α of 0.90.

#### 2.2.4. Edinburgh Postpartum Depression Scale (EPDS)

The EPDS was developed by Cox et al. [42] and translated by Lee et al. [43]. In this study, the Chinese version of the EPDS was used to assess postpartum depression symptoms. The 4-point Likert scale comprises 10 items, with ratings from 1 to 4. The total score ranges from 0 to 30. A score of 13 or higher indicates depression, with higher scores denoting more severe depression. In this study, Cronbach’s α of the Chinese version of EPDS was 0.90, indicating satisfactory reliability.

#### 2.2.5. Dutch Eating Behavior Questionnaire (DEBQ)

The DEBQ, developed by van Strien [9], is a self-report questionnaire comprising three subscales that measure emotional, external, and restrained eating styles. The instrument features 33 items divided into emotional eating (13 items), external eating (10 items), and restrained eating (10 items). The responses are recorded on a 5-point scale, ranging from 1 (‘never’) to 5 (‘very often’), to determine the frequency of engagement in specific eating styles. Subscales are calculated by averaging the corresponding items, with higher scores indicating more prevalent eating issues [9]. In this study, Cronbach’s α for emotional eating, external eating, and restrained eating were 0.96, 0.89, and 0.93, respectively.

### 2.3. Data Analysis 

#### 2.3.1. Descriptive Analysis

SPSS software, version 26.0, was used for statistical analysis. Count data were reported as frequencies and percentages. The measurement data were presented as mean ± standard deviation (M ± SD). 

#### 2.3.2. Latent Profile Analysis

Mplus software, version 8.3, was used to analyze the latent profiles of emotional eating, external eating, and restrained eating components of the DEBQ. The fitting indices include the Akaike Information Criterion (AIC), Bayesian Information Criterion (BIC), adjusted Bayesian Information Criterion (aBIC), Entropy, Lo-Mendell-Rubin (LMR), and Bootstrapped Likelihood Ratio Test (BLRT). Generally, smaller statistical values indicate a better model fit. Entropy measures classification accuracy; values closer to 1 indicate more accurate classification. LMR and BLRT, corresponding to *p* < 0.05, were used to compare the fit differences between the k-1 and k-category models. In addition to these indicators, practical significance and clinical interpretability should also be considered.

#### 2.3.3. Single-Factor and Multi-Factor Analysis

Based on the specified model, the Welch test and chi-square test were used for between-group comparisons. Indicators that reached statistical significance underwent multinomial logistic regression analysis to examine the factors influencing different profiles of eating styles among postpartum women. *p* < 0.05 indicated that the difference was statistically significant.

### 2.4. Ethical Considerations

This study was conducted according to the guidelines of the Declaration of Helsinki and approved by the Ethics Committee of the School of Nursing, Central South University, with the approval number E2021108 (29 July 2021). Before the study, both oral and written consents were obtained from all eligible participants, ensuring that participation or non-participation would not affect their work performance. Additionally, all collected information would be anonymous and de-identified. Furthermore, participants were informed that they could withdraw from the study at any time.

## 3. Result

### 3.1. Sample Characteristics

In this study, 507 valid questionnaires were collected. The average age of the postpartum women was 30.92 years (SD = 4.67, range 19–48). Most of the surveyed women lived in urban areas (*n* = 346, 68.2%), had a university degree (*n* = 358, 70.6%), were incumbent (*n* = 327, 64.5%), had a monthly income up to RMB 5000 (*n* = 344, 67.9%), and experienced a general sleep condition. The average BMI of the participants was 22.94 (SD = 3.00), and the PPWR was 4.72 (SD = 6.30). The average score of PWSQ, WBIS, and EPDS were 0.46 (SD = 1.23), 26.9 (SD = 9.4), and 8.2 (SD = 6.0), respectively. Details are shown in Table 1.

### 3.2. Results of Latent Profile Analysis

An initial analysis of 1–5 clusters using the scores from the emotional eating, external eating, and restrained eating components of the DEBQ scale was conducted. The results of the model-fit metrics are in Table 2, where AIC, BIC, and aBIC gradually decreased. The AIC was the same as in the four- and five-profile models. The BLRT *p*-value was consistently <0.05 across all potential profile models, while the LMR *p*-value was > 0.05. Entropy was highest in the five-profile model. The five-profile model had an LMR *p*-value > 0.05, so it was excluded. The four-profile model seemed to be the fittest. But, its two subgroups accounted for only a few individuals, and its clinical interpretability was poor. Considering the above reasons, the three-profile model was chosen as the optimal model. Table 3 displays the attribution probability matrix for the three potential profiles. Each class’s average probability of attribution to its corresponding profile ranged from 85.3% to 92.1%, indicating that the model results for the three potential profiles were reasonable.

### 3.3. Categories of Latent Profile

Based on the latent profile analysis results, the scores of the three profiles on the three dimensions of the DEBQ are illustrated in Figure 1. 

Profile 1 comprised postpartum women with low levels of emotional, external, and restrained eating (6.9% of the sample). Women in this profile scored significantly lower than those in Profiles 2 and 3 on each dimension; this profile was named ‘Low-level’.

Profile 2 included postpartum women with medium levels of emotional, external, and restrained eating (66.1% of the sample). Women from this profile scored between Profiles 1 and 3 on each dimension, and this profile was named ‘Medium-level’.

Profile 3 consisted of postpartum women with high levels of emotional, external, and restrained eating (27.0% of the sample). Women from this profile scored significantly higher than Profiles 1 and 2 on each dimension, and this profile was named ‘High-level’.

### 3.4. Inter-Profile Characteristic Differences

This LPA solution demonstrated that the scores of the three DEBQ components significantly contributed to the overall solution (Table 4), confirming the validity of the latent profile analysis grouping by a one-way ANOVA.

Differences in demographic characteristics, eating styles, PWS, WBIS, EPDS, and PPWR across profiles were compared in Table 4. The results indicated that BMI, PWS, WBIS, EPDS, PPWR, and sleep condition differed significantly (*p* < 0.05). No significant differences were found in the remaining variables (*p* > 0.05). 

### 3.5. Multinomial Logistic Regression of Eating Styles Profiles

In this study, three profiles were used as the dependent variable. Variables that were statistically significant in the univariate analysis were used as independent variables. A very good sleep condition was used as the reference category. Correlations were explored through multinomial logistic regression. Figure 2 indicates that the WBIS, EPDS, BMI, and PPWR were significantly associated with the profiles (*p* < 0.05). 

In general, comparisons between Profile 1 and Profile 2 eating styles among postpartum women indicated that EPDS (OR = 1.276, 95%CI: 1.147–1.420) and BMI (OR = 1.221, 95%CI: 1.022–1.459) were more likely to be associated with Profile 2, whereas PPWR (OR = 0.863, 95%CI: 0.792–0.942) was more likely to be associated with Profile 1. When comparing Profile 3 and Profile 1 of disordered eating behaviors among postpartum women, EPDS (OR = 1.334, 95%CI: 1.194–1.489) and BMI (OR = 1.360, 95%CI: 1.126–1.642) were more likely to be associated with Profile 3, while PPWR (OR = 0.886, 95%CI: 0.809–0.970) was more likely to be associated with Profile 1. Using Profile 2 as a reference, EPDS (OR = 1.045, 95%CI: 1.003–1.089) and BMI (OR = 1.114, 95%CI: 1.024–1.212) were more likely to be associated with Profile 3 of eating styles among postpartum women.

## 4. Discussion

The present study focused on establishing eating styles profiles of postpartum women based on their disordered eating behaviors (emotional eating, external eating, and restrained eating) and explored the correlates. The results showed that the presence of the three disordered eating behaviors can be heterogeneous in postpartum women with three profiles: postpartum women with low levels of emotional, external, and restrained eating (Profile 1, 6.9%); postpartum women with medium levels of emotional, external, and restrained eating (Profile 2, 66.1%); and postpartum women with high levels of emotional, external, and restrained eating (Profile 3, 27.0%).

The present study found that 93.1% (Profile 2 and Profile 3) of Chinese postpartum women exhibited medium to high levels of disordered eating behaviors, significantly higher than the 58.49% (medium and high level) reported among the general population of women in Chile [36]. This can be attributed to a combination of physiological, cultural, and psychological factors unique to the postpartum period. Physiological changes during the postpartum period led to natural weight fluctuations, heightened concerns about body image, and weight retention [16,18]. The traditional Chinese custom of ‘confinement in childbirth’, characterized by reduced physical activity and consumption of nutritious soups, exacerbates these weight management challenges [17]. Furthermore, the transition to motherhood involves disruptions in routine and sleep, complicating the maintenance of regular eating patterns [19]. There is also a noted decline in motivation to maintain a healthy lifestyle postpartum compared to during pregnancy [18,20]. These factors, coupled with the psychological, social, and economic stressors associated with motherhood, distort perceptions of body image and weight and increase the risk of disordered eating behaviors [21]. This disparity underscores the need for more focused research on the unique eating-behavior challenges faced by postpartum women, highlighting the importance of addressing their specific needs to reduce the prevalence of disordered eating behaviors in this high-risk group. 

Additionally, this study demonstrated that BMI, PPWR, and PPD were associated with eating styles profiles in postpartum women. Postpartum women with a higher BMI were more likely to be classified into either of the two higher-level disordered eating-behavior profiles (Profile 2 and Profile 3). This aligns with previous research showing that individuals with a higher BMI are at a higher risk for disordered eating behaviors [44,45]. In recent years, ‘thinness as beauty’ has become increasingly popular in China [46]. The whole sociocultural environment is characterized by a low tolerance for deviation from the ‘thin idealized’. Meanwhile, there is a deep-rooted collectivist culture in China, which emphasizes close ties and consistency between individuals and groups. Heightened sensitivity to group norms, characteristic of collectivist groups, may also contribute to increased stress [47]. Therefore, this thin idealized sociocultural pressure may lead to individuals being dissatisfied with their body image, which in turn, leads to greater disordered eating behaviors [48]. It is worth noting that the mean BMI of Profile 3 was higher than Profiles 1 and 2, and the difference was statistically significant. Meanwhile, the mean BMI difference between Profiles 1 and 2 was not statistically significant. This suggests that, while postpartum women with a higher BMI were more likely to be at greater levels of disordered eating behaviors, those with a lower BMI were also likely to be at moderate levels of disordered eating behaviors. Thus, the present findings suggest that efforts seeking to improve postpartum women’s eating behaviors should not be limited to those with a high BMI.

Surprisingly, this study found that, compared to Profile 1, the higher PPWR was less likely to fall into Profiles 2 and 3. This means that PPWR is a protective factor for disordered eating behaviors in the postpartum period. Yet, studies among general populations clearly show positive cross-sectional associations between disordered eating behaviors and weight gain [49]. This may be related to the specific psychological and behavioral patterns of postpartum women. On the one hand, postpartum women may face pressure from social and family expectations that they can quickly return to pre-pregnancy weight. This pressure may lead them to adopt restrained eating control measures to facilitate weight recovery [50]. While such pressure may contribute to the control of eating behaviors in the short term, it may have a negative impact in the long term. 

On the other hand, from China’s particular socio-cultural perspective, the rationality and functionality of PPWR during ‘confinement in childbirth’ or breastfeeding is also a probable explanation. ‘Confinement in childbirth’ is a traditional Chinese custom that emphasizes the need for postpartum women to increase their intake of nutrient-rich foods and have a large body to meet the double needs in restoring health and breastfeeding [51]. PPWR is considered reasonable in this particular cultural context, which helps postpartum women maintain healthy eating behaviors. Furthermore, in China, where family ties are particularly strong, family responsibilities and duties are deeply embedded in sociocultural beliefs and hold significant importance [52]. Failure to fulfill these responsibilities often results in feelings of guilt [53]. Chinese families typically prioritize the best interests of their children. Consequently, Chinese postpartum women may perceive the effective nurturing of infants as a crucial family responsibility. This perception may lead them to deprioritize other personal needs, such as returning to pre-pregnancy physical shape, while prioritizing physical health and breastfeeding. When PPWR is associated with the restoration of physical health or improved breastfeeding outcomes, postpartum women may feel empowered and valued as mothers. This recognition and assumption of family responsibilities, coupled with increased self-efficacy, may facilitate better management of their eating behaviors [54].

It is noteworthy that previous ‘variable-centered’ studies found no association between PPWR and restricted eating [49,55]. In this study also, no association was found between PPWR and restricted eating when comparing Profile 3 with Profile 2. This indirectly illustrates the heterogeneity of eating behaviors in postpartum women and the specificity of conducting studies from a ‘person-centered’ perspective. In addition, no previous studies have reported an association between PPWR and external and emotional eating [49]. These findings not only provide new perspectives but also offer valuable insights for future research.

Consistent with the previous literature on the general population [18,26], our results showed that depression symptoms are also a risk factor for disordered eating behaviors in postpartum women. The postpartum period is a vulnerable time for the development of depression [56], and as a specific psychological stressor, its impact on eating behaviors should not be ignored. Meanwhile, disordered eating behaviors may further exacerbate PPD symptoms [57], forming a vicious cycle. The results also showed that PPD affects eating behaviors at two extremes. We underscore, in particular, that PPD not only has positive correlations with emotional and external eating but also is positively associated with restrained eating. Interventions seeking to address disordered eating behaviors in postpartum women and mitigate the impact of these high-risk eating styles on the next generation may benefit from treating PPD.

Contrary to predictions, and somewhat puzzlingly, the present study has not found an association between internalized weight stigma and disordered eating behaviors. This is inconsistent with findings in the general population that internalized weight stigma was positively associated with disordered eating behaviors [58]. Internalized weight stigma involves weight-based self-devaluation [59], which is negatively correlated with dietary self-efficacy [60], and decreased eating self-efficacy may lead to a decline in the ability of individuals to control their eating behaviors. The possible explanations for this discrepancy in the study results are the relationship between the developmental mechanisms of disordered eating behaviors and the internalized weight stigma was weak or more complex. Moreover, it is possible that disordered eating behaviors are more related to other factors, such as life stressors, emotional regulation skills, or social support, rather than being driven primarily by an individual’s self-devaluation based on weight. Future research needs to more thoroughly understand and explore the developmental mechanisms of disordered eating behaviors in postpartum women to provide more comprehensive and individualized interventions.

Furthermore, this study also did not find links between perceived weight stigma and disordered eating behaviors in postpartum women. Zancu and Diaconu-Gherasim [61] conducted a path analysis on Romanian adolescents, revealing that perceived weight stigma was not directly linked to disordered eating behaviors but rather indirectly through internalized weight stigma and body esteem. This indicates that, similar to the general population, the relationship between perceived weight stigma and disordered eating behaviors in postpartum women is complex and mediated by multiple factors. Further research is necessary to explore the underlying mechanisms and develop more effective intervention strategies to support postpartum women in maintaining healthy eating behaviors.

## 5. Limitations

This study has several limitations. First, all measures were self-reported, excluding weight and height. However, these scales have all shown good psychometric properties in this study, ensuring the accuracy of the results. Second, our sample was limited to one city. Future researchers should consider extending the research to multiple regions to improve sample representativeness, thereby accurately estimating the prevalence of weight stigma and its correlates in postpartum women. 

## 6. Implications

This study marks the first exploration of disordered eating behaviors among postpartum women using a ‘person-centered’ approach. The prevalence of moderate to high levels of disordered eating behaviors underscores the urgent need for targeted interventions within postpartum care. The identification of diverse traits highlights the heterogeneity of eating’s behavioral manifestations during this critical life stage. Moreover, the observed associations between relevant factors and eating styles profiles illuminate the complex interplay of the sociocultural, psychological, and physiological factors influencing postpartum women’s eating behaviors. 

In particular, the study highlights the impact of Chinese traditions and psychosocial cultural beliefs on disordered eating behaviors among postpartum women. The cultural emphasis on thinness can lead to increased stress and disordered eating behaviors. However, traditional practices, such as ‘confinement in childbirth’, family ties, and family responsibility in Chinese culture, may provide a protective effect against disordered eating behaviors. Clinicians must consider these cultural factors and the unique lifestyle practices of the local population when developing appropriate strategies. These strategies should include education for mothers and their family members about healthy eating behaviors, the importance of maternal mental health, and managing sociocultural pressures. It is crucial to educate family members to recognize and affirm the mother’s role and responsibilities within the family, thereby providing emotional support and validation. Postpartum follow-up should be emphasized to provide continuous support, helping mothers maintain healthy practices. Additionally, strategies to enhance the mother’s self-efficacy, such as setting achievable health goals and providing positive reinforcement, may empower mothers and improve their eating behaviors.

The findings also emphasize the importance of addressing maternal mental health, particularly PPD. Interventions aimed at mitigating PPD symptoms may have dual benefits in improving both maternal mental health and eating behaviors. In conclusion, this study contributes to the global understanding of disordered eating behaviors in postpartum women, highlighting the need for comprehensive culturally sensitive interventions that address the multifaceted determinants of these behaviors to promote maternal health.

## 7. Conclusions

This study revealed that disordered eating behaviors in postpartum women are prevalent and can be heterogeneous with three profiles: low levels of emotional, external, and restrained eating (Profile 1); medium levels (Profile 2); and high levels (Profile 3). BMI, PPWR, and PPD were associated with the eating styles profiles of postpartum women. This evidence lays an important foundation for guiding healthcare professionals in developing targeted interventions based on the identified characteristics, with the ultimate goal of improving maternal and child health globally.

## Figures and Tables

**Figure 1 nutrients-16-02299-f001:**
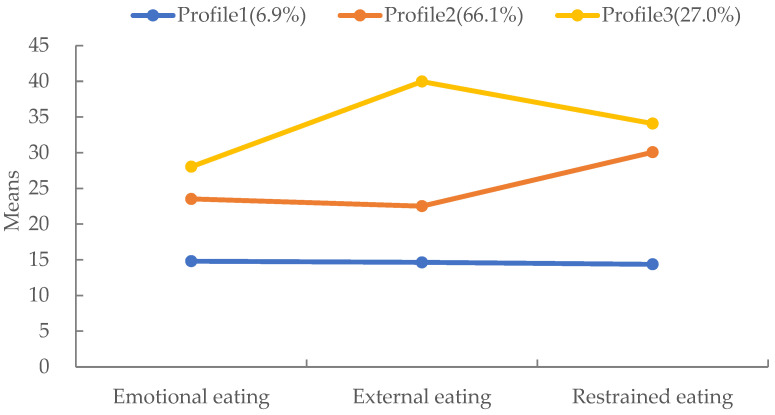
Latent profile model of eating styles in postpartum women.

**Figure 2 nutrients-16-02299-f002:**
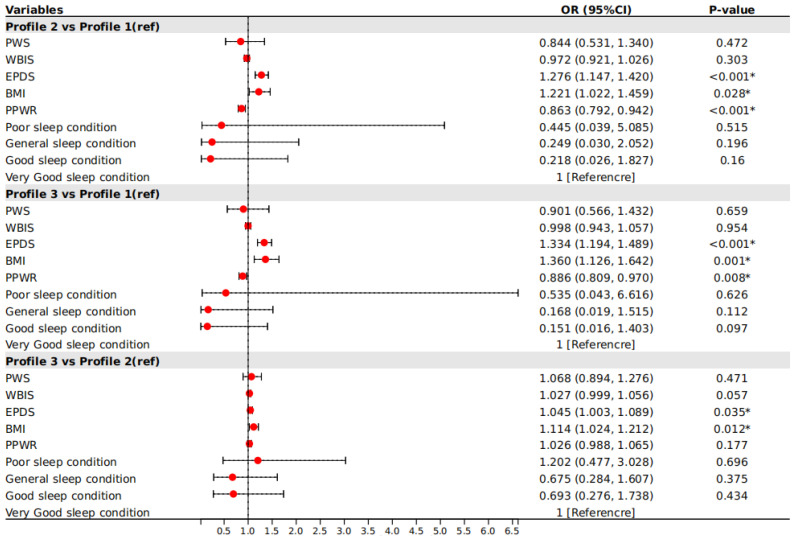
Multinomial logistic regression of eating styles profiles. Note: BMI = body mass index; PPWR = postpartum weight retention; PWSQ = Perceived Weight Stigma Questionnaire; WBIS = Weight Bias Internalization Scale; EPDS = Edinburgh Postpartum Depression Scale. * Statistically significant. The solid red circle indicates the effect value of a single study, and the horizontal lines across the ends of the solid red circle indicate the confidence intervals; the longer the line, the larger the confidence interval and the lower the confidence in the findings.

**Table 1 nutrients-16-02299-t001:** General characteristics of postpartum women (N = 507).

Variables	Categories	N (%)	Mean (SD)
Age	_	_	30.92 (4.67)
Residence	Rural	73 (14.4)	_
	Urban	346 (68.2)	_
	Suburb	88 (17.4)	_
Education level	High school and below	71 (14.0)	_
	University	358 (70.6)	_
	Postgraduate and above	78 (15.4)	_
Monthly income (RMB)	<3000	44 (8.7)	_
	3000~5000	119 (23.5)	_
	>5000	344 (67.9)	_
Employment status	Unemployment	59 (11.6)	_
	Incumbent	327 (64.5)	_
	Liberal profession	121 (23.9)	_
Sleep condition	Poor	104 (20.5)	_
	General	242 (47.7)	_
	Good	126 (24.9)	_
	Very good	35 (6.9)	_
BMI	_	_	22.94 (3.00)
PPWR	_	_	4.72 (6.30)
PWSQ	_	_	0.46 (1.23)
WBIS	_	_	25.50 (9.26)
EPDS	_	_	8.26 (5.91)
Emotional eating	_	_	27.61 (10.89)
External eating	_	_	29.93 (7.14)
Restrained eating	_	_	24.07 (8.73)

Note: BMI = body mass index; PPWR = postpartum weight retention; PWSQ = Perceived Weight Stigma Questionnaire; WBIS = Weight Bias Internalization Scale; EPDS = Edinburgh Postpartum Depression Scale. ‘_ ’ indicates no data.

**Table 2 nutrients-16-02299-t002:** Indicators for each latent profile of eating styles among postpartum women.

Model	AIC	BIC	aBIC	*P*LMR	*P*BLRT	Entropy	Group Size for Each Profile				
							1	2	3	4	5
1-Class	10,936.015	10,961.386	10,942.341	_	_	_	507				
2-Class	10,799.329	10,841.614	10,809.873	0.0000	0.0000	0.854	48	459			
**3-Class**	**10,698.627**	**10,757.826**	**10,713.388**	**0.0023**	**0.0000**	**0.779**	**335**	**35**	**137**		
4-Class	10,650.851	10,726.964	10,669.830	0.0045	0.0000	0.801	278	35	168	26	
5-Class	10,578.324	10,671.351	10,601.520	0.0522	0.0000	0.879	37	163	170	111	26

Note: Bold entries reflect the selected model. Class refers to the number of potential categories, with 1-Class to 5-Class meaning that it is divided into 1–5 potential categories. Through model comparison, the optimal number of categories is determined. ‘_ ’ indicates no data.

**Table 3 nutrients-16-02299-t003:** Indicators for each latent profile of eating styles in postpartum women.

3-Class	Profile 1	Profile 2	Profile 3
Profile 1	**0.921**	0.008	0.061
Profile 2	0.178	**0.920**	0.000
Profile 3	0.145	0.000	**0.853**

Note: 3-Class means that it is divided into 3 potential categories.

**Table 4 nutrients-16-02299-t004:** Inter-profile characteristic differences (N, %).

Variable	Profile 1	Profile 2	Profile 3	*F/*χ^2^	*p*
Emotional eating (M ± SD)	14.17 ± 2.80	22.06 ± 6.18 ^a^	40.93 ± 6.92 ^ab^	521.992	<0.001 *
External eating (M ± SD)	13.06 ± 3.63	30.00 ± 5.41 ^a^	34.07 ± 4.86 ^ab^	230.814	<0.001 *
Restrained eating (M ± SD)	14.54 ± 6.24	23.36 ± 8.08 ^a^	28.21 ± 8.45 ^ab^	43.658	<0.001 *
Age (M ± SD)	30.80 ± 5.42	30.82 ± 4.44	31.17 ± 5.01	0.275	0.760
BMI (M ± SD)	22.69 ± 2.73	22.58 ± 2.86	23.91 ± 3.19 ^ab^	10.109	<0.001 *
PWSQ (M ± SD)	0.31 ± 1.21	0.36 ± 1.07	0.74 ± 1.52 ^b^	5.209	0.006 *
WBIS (M ± SD)	24.06 ± 9.63	24.21 ± 8.57	29.01 ± 9.91 ^ab^	14.240	<0.001 *
EPDS (M ± SD)	3.34 ± 3.97	7.96 ± 5.51 ^a^	10.26 ± 6.40 ^ab^	22.101	<0.001 *
PPWR (M ± SD)	7.47 ± 6.71	3.92 ± 6.10 ^a^	5.95 ±6.33 ^b^	8.885	<0.001 *
Residence				1.396	0.845
Rural	6 (17.1)	51 (15.2)	16 (11.7)		
Urban	24 (68.6)	226 (67.5)	96 (70.1)		
Suburb	5 (14.3)	58 (17.3)	25 (18.2)		
Education level				7.785	0.100
High school and below	6 (17.1)	48 (14.3)	17 (12.4)		
University	27 (77.1)	241 (71.9)	90 (65.7)		
Postgraduate and above	2 (5.7)	46 (13.7)	30 (21.9)		
Monthly income (RMB)				4.747	0.314
<3000	5 (14.3)	29 (8.7)	10 (7.3)		
3000~5000	10 (28.6)	83 (24.8)	26 (19.0)		
>5000	20 (57.1)	223 (66.6)	101 (73.7)		
Employment status				6.593	0.159
Unemployment	4 (11.4)	36 (10.7)	19 (13.9)		
Incumbent	17 (48.6)	221 (66.0)	89 (65.0)		
Liberal profession	14 (40.0)	78 (23.3)	29 (21.2)		
Sleep condition				15.116	0.019 *
Poor	3 (8.6)	60 (17.9)	41 (29.9) ^ab^		
General	18 (51.4)	163 (48.7)	61 (44.5)		
Good	13 (37.1)	87 (26.0)	26 (19.0)		
Very good	1 (2.9)	25 (7.5)	9 (6.6)		

Note: ^a^ indicates that, compared with Profile 1, *p* < 0.05; ^b^ indicates that compared with Profile 2, *p* < 0.05. Variance analysis was performed using the Welch method. BMI, WBIS, PPWR, external eating, and restrained eating demonstrated variance homogeneity, and the LSD test was used for multiple comparisons. For the remaining continuous variables, which exhibited variance non-homogeneity, Dunnett’s T3 method was used for multiple comparisons. * Statistically significant.

## Data Availability

The data presented in this study are available upon request from the corresponding author. The data are not publicly available due to reasons of privacy.

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
