# Peer review of "Eating Styles Profiles and Correlates in Chinese Postpartum Women: A Latent Profile Analysis"

_nutrients, 2024, doi:10.3390/nu16142299_

Round 1

Reviewer 1 Report

Comments and Suggestions for Authors

The authors conducted an interesting study on the dietary habits in Chinese women. The methodology and discussion can made clearer and enriched if the following are addressed:

1. Can the authors explain the 3 types of eating disorders? How is putting retrained eating together with emotional and external eating explained the various profiles? This is assuming that a woman with high levels of restrained eating should not have high level of emotional or external eating or are there clear associations that if one type of disordered eating is high, the others are also high levels? This needs further explanation to understand why the authors conclude on using these 3 chosen profiles. This is also fundamental as the results and conclusions are based on this model.

2. I would like to see more on the impact of the Chinese traditions and psychosocial cultural beliefs which may directly impact the disordered in postpartum mothers. Clinicians have to take this into account based on the local population unique lifestyle practices and beliefs; this will then translating to the appropriate strategies which includes education of the mother and her family members antenatally and post partum follow up.

It is important that the above points are fully addressed.

Reviewer 2 Report

Comments and Suggestions for Authors

Dear Authors

Thank you for the opportunity to review this paper.

The paper is well written and the statistical tests are robust.

Please find my comments below:

Results section

Please put Table 2 in section 3.2  Results of latent profile analyses.

Please explain Table 2 and 3 what does it mean class1, class2, etc

Discussion

At the beginning of this section please repeat the aim of the study.

Kind regards
